# Rotavirus Infections: Pathophysiology, Symptoms, and Vaccination

**DOI:** 10.3390/pathogens14050480

**Published:** 2025-05-14

**Authors:** Karolina Pawłuszkiewicz, Piotr Józef Ryglowski, Natalia Idzik, Katarzyna Błaszczyszyn, Emilia Kucharczyk, Dagmara Gaweł-Dąbrowska, Marta Siczek, Jarosław Widelski, Emil Paluch

**Affiliations:** 1Faculty of Medicine, Wroclaw Medical University, Wybrzeże L. Pasteura 1, 50-367 Wrocław, Poland; karolina.pawluszkiewicz@student.umw.edu.pl (K.P.); piotr.ryglowski@student.umw.edu.pl (P.J.R.); natalia.idzik@student.umw.edu.pl (N.I.); emilia.kucharczyk@student.umw.edu.pl (E.K.); 2Jan Mikulicz-Radecki University Hospital in Wroclaw, Borowska 213, 50-556 Wrocław, Poland; kb.blaszczyszyn@gmail.com; 3Division of Public Health, Faculty of Health Sciences, Wroclaw Medical University, Bujwida 44, 50-345 Wroclaw, Poland; dagmara.gawel-dabrowska@umw.edu.pl; 4Department of Forensic Medicine, Wroclaw Medical University, J. Mikulicza-Radeckiego 4, 50-345 Wroclaw, Poland; marta.siczek@umw.edu.pl; 5Department of Pharmacognosy with Medicinal Plant Unit, Medical University of Lublin, Chodźki 1, 20-093 Lublin, Poland; 6Department of Microbiology, Faculty of Medicine, Wroclaw Medical University, St. T. Chałubińskiego 4, 50-376 Wrocław, Poland

**Keywords:** rotavirus, diarrhea, pathogenesis, vaccination, infants

## Abstract

Rotavirus (RV) is the most common cause of severe acute gastroenteritis (AGE) in children under five years of age. This review summarizes current knowledge on RV infections, with a particular focus on viral structure, pathophysiological mechanisms, and age-dependent clinical presentation. Special attention is given to systemic manifestations, including central nervous system involvement, autoimmune responses such as type 1 diabetes and celiac disease, and rare associations with biliary atresia. The mechanisms of RV-induced diarrhea and vomiting are discussed in detail. Clinical severity scoring systems—such as the Vesikari and Clark scales—and dehydration assessment tools—the Clinical Dehydration Scale (CDS) and the Dehydration: Assessing Kids Accurately (DHAKA) score—are compared. The review highlights differences in disease course between children under and over five years, emphasizing that RV is not limited to early childhood. A major section addresses the global effectiveness of vaccination programs, their role in reducing disease burden, coverage challenges, and decreased efficacy in low-income countries. Particular focus is placed on high-risk groups, including preterm and immunocompromised infants.

## 1. Introduction

Rotaviruses (RVs) are a highly contagious group of viruses that are the leading cause of severe, dehydrating diarrhea in children under the age of five worldwide [1]. Prior to the introduction of vaccines, RV infections were responsible for up to 500,000 child deaths annually. They accounted for 30% to 50% of all hospitalizations related to gastroenteritis in children under five years old [2].

In the literature review [3] examining the impact of RV vaccination, which included 101 studies from 47 countries across different child mortality levels, median relative reductions were observed of 59% in RV-related hospitalizations and approximately 36% in hospitalizations and mortality due to acute gastroenteritis (AGE) among children under five following the introduction of the RV vaccine. Importantly, reductions in AGE and RV-specific hospitalizations decreased progressively over time after the vaccine’s introduction, with the most significant decreases observed in regions with the highest RV vaccine coverage [3]. Between 2006 and 2019, RV vaccines were credited with preventing approximately 140,000 child deaths. Additionally, relaxing age restrictions on these vaccines prevented up to 17,000 deaths from 2013 to 2019. In 2019 alone, RV vaccines prevented 15% of under-five RV deaths (0.5% of child mortality) [4]. According to the Global Burden of Disease Study published in 2019, RV still causes an estimated 235,331 deaths annually [5].

The primary site of RV A infection is the gastrointestinal tract, with the virus being excreted in the feces of infected individuals for a period of 5 to 7 days. Transmission primarily occurs via the fecal-oral route [6]. The virus can also spread easily through hand-to-hand contact, increasing the likelihood of infection. Although the likelihood of zoonotic transmission of RV from animals to humans remains low, such events are generally not associated with significant clinical manifestations [1]. RVs are highly resistant to environmental factors, including temperature and pH, which enhances their infectious potential [6].

Recent studies have indicated that mouse murine RV strains can replicate not only in the intestinal epithelium but also in the salivary glands of both pups and their mothers, allowing transmission through saliva. Orally infected puppies are capable of transmitting the virus to their mothers through suckling, thereby leading to infection of the mammary gland. In infected adult mice, the virus was detected in saliva as early as 3 days post-infection, with viral genomes identified up to 3 weeks post-infection, depending on the strain [6,7]. A study on Gn piglets revealed the presence of the virus in the ileum, nasal cavity, and salivary glands two days after inoculation with the Rotarix vaccine, the attenuated Wa RV strain (Wa AttHRV), or recombinant rhesus RVs (rRRV). The duration of viral replication and shedding in the salivary glands remains undetermined, as the animals were euthanized during the early stages of the study [7,8].

This review presents a comprehensive overview of RV infections, focusing not only on clinical manifestations but also on the underlying pathophysiological mechanisms and viral structure. It discusses the pathophysiology of RV-related symptoms and their clinical presentation, with a particular emphasis on age-related differences in children under and over five years of age. Additionally, the review includes detailed information on current vaccination strategies, the impact of immunization programs on disease burden, and their role in preventing both local and systemic complications. By integrating recent findings on RV pathogenesis, clinical patterns, and prevention, this review aims to support the development of more effective strategies for the management and control of RV infections in the pediatric population.

## 2. Materials and Methods

We conducted a comprehensive analysis of RV infections. While the primary focus was on the literature published within the past decade, the review spans a timeline from 1978 to 2024 to ensure a comprehensive, thorough perspective. Key search terms included “rotavirus structure”, “rotavirus pathophysiology”, “rotavirus infections”, “rotavirus infection in children under 5”, “rotavirus systemic infections”, “rotavirus CNS”, “rotavirus biliary atresia”, and “rotavirus vaccine”. Figures 1 and 2 were created using draw.io v27.0.5 (JGraph Ltd., Northampton, UK; draw.io AG, Zürich, Switzerland; operational headquarters: Wiesbaden, Germany). Figure 3 was generated using Python 3.12.3 (Python Software Foundation, Wilmington, DE, USA). We describe both local and systemic manifestations, with particular emphasis on the underlying pathophysiological mechanisms and viral structure. Our review emphasizes clinical presentation across pediatric age groups, including distinctions between children under and over five years of age. Furthermore, we present an in-depth evaluation of RV vaccination, highlighting its effectiveness, global implementation, and impact on disease burden, with special attention to vulnerable populations such as immunocompromised children and preterm infants.

## 3. RVs Structure

RVs are double-stranded RNA (dsRNA) viruses classified within the genus *Rotavirus* of the family *Sedoreoviridae*. Their genome consists of 11 segments of dsRNA, comprising approximately 18,500 base pairs in total. These segments encode six structural proteins (VP1–VP4, VP6, and VP7) and six non-structural proteins (NSP1–NSP6). While the majority of genome segments are monocistronic, segment 11 is an exception in some strains, as it contains two overlapping open reading frames (ORFs) that allow simultaneous expression of NSP5 and NSP6 [9].

As shown in Figure 1, the mature, infectious RV virion consists of three concentric protein layers that encapsulate the segmented dsRNA genome. The innermost core comprises 120 VP2 molecules arranged into twelve asymmetric decamers. Associated with the five-fold symmetry axes of this core are complexes containing the RNA-dependent RNA polymerase (VP1) and the capping enzyme (VP3). The intermediate layer is composed of VP6, while the outermost layer includes the glycoprotein VP7 and the spike protein VP4, which extends outward from the virion surface [10,11].

Structural (VP) and non-structural (NSP) RV proteins play essential roles in viral replication, the manifestation of clinical symptomatology, and the classification of RV species, as summarized in Table 1 [10,11,12,13,14,15,16,17].

RVs are classified into distinct species based on antibody reactivity and the sequence identity of the VP6 protein. The recognized species of RVs include A, B, C, D, E, F, G, H, I, and J [18]. The most common one is RV A, which is responsible for 90% of the infection cases [19]. RV A is the leading cause of infection in infants and young children, but it also affects various mammals and birds. RV B is frequently detected in diarrheic pigs and has been implicated in sporadic outbreaks of diarrhea in humans [19]. Other species of RVs are primarily detected in animals: H—pigs [20], D, F, G—birds [21], I—cats [22], and J—bats [23]. For RV A, a genotyping system targeting 11 genome segments reveals extensive diversity of this virus: 42 G-types (for glycosylated protein VP7) and 58 P-types (for proteolytically cleaved protein VP4) [18].

## 4. Local and Systemic Symptoms Caused by RV

The primary site of RV A infection is the gastrointestinal tract, with the virus being shed in the feces of infected individuals for a period of 5 to 7 days [24]. Transmission primarily occurs via the fecal-oral route [6]. Direct contact—particularly hand-to-hand—may also facilitate spread. Although the risk of zoonotic transmission of RV from animals to humans is low, such transmissions are unlikely to cause significant clinical illness. RVs exhibit substantial resistance to environmental stressors, including temperature and pH, which enhances their infectious potential [24]. RV infection primarily manifests as AGE, characterized by diarrhea and vomiting, which can lead to systemic complications such as dehydration. Additionally, recent studies indicate the potential role of RV in central nervous system complications, autoimmune disorders, biliary atresia, and respiratory diseases.

### 4.1. Diarrhea, Vomiting, and Dehydration

The American Academy of Pediatrics (AAP) characterizes AGE as a diarrheal illness with a rapid onset, which may be accompanied by symptoms such as nausea, vomiting, fever, or abdominal pain, though these symptoms are not mandatory for diagnosis [8]. The World Health Organization (WHO) defines acute diarrhea as the excretion of three or more loose or liquid stools per day for a duration of 3 to 13 days [25,26].

Diarrhea in RV A infection is non-bloody and associated with a limited inflammatory response [27]. Inflammatory markers such as serum amyloid A (SAA), C-reactive protein (CRP), and white blood cell (WBC) counts in RV diarrhea are significantly lower than in bacterial infections; thus, RV diarrhea is considered non-inflammatory. Despite low results, the diagnostic efficacy of SAA, CRP, and WBC is still indicated [28].

Diarrhea is one of the most common symptoms of RV infection and is caused by several mechanisms. As shown in Figure 2, these mechanisms include secretory, osmotic, and neurogenic pathways. One of the mechanisms responsible for secretory diarrhea is the viral enterotoxin NSP4, which, via the phospholipase C signaling route, leads to the activation of chloride channels and the secretion of these ions into the intestinal lumen. The resultant osmotic gradient promotes the movement of water into the intestinal lumen, thereby contributing to the development of diarrhea [29].

The viral enterotoxin NSP4 and the replication of the virus within the cell disrupt cellular homeostasis [29,30]: shortening and loss of villi, loss of microvilli, infiltration of mononuclear cells, expansion of the endoplasmic reticulum, and swelling of mitochondria in enterocytes [31]. Proposed mechanisms for these effects include virus-induced apoptosis [32], NSP4-dependent mislocalization of the tight junction protein ZO-1 [33], and binding to extracellular matrix proteins of the basement membrane, such as the laminin-β3 subunit and fibronectin [34]. These mechanisms lead to impaired absorption of nutrients and electrolytes, resulting in osmotic diarrhea.

The viral enterotoxin also stimulates the release of 5-hydroxytryptamine (5-HT), known as serotonin, in a calcium-dependent signaling pathway. 5-HT3 activates the enteric nerves innervating the small intestine, increasing intestinal motility and leading to diarrhea [35].

While serotonin secretion stimulates vagus nerve signaling between the gut and brain—contributing to nausea and vomiting (supporting the potential use of 5-HT3 antagonists like ondansetron), murine knockout models have demonstrated a more complex mechanism. This study demonstrates that the 5-HT3 receptor mediates RV-induced intestinal dysmotility but not central emetic signaling, suggesting the involvement of an additional neural pathway [36,37].

Vomiting, which usually occurs early in the disease, can also be caused by delayed gastric emptying, although it is unclear whether this is due to vagus nerve activation during RV infections or increased levels of gastrointestinal hormones (e.g., secretin, gastrin, or cholecystokinin) [38].

**Figure 2 pathogens-14-00480-f002:**
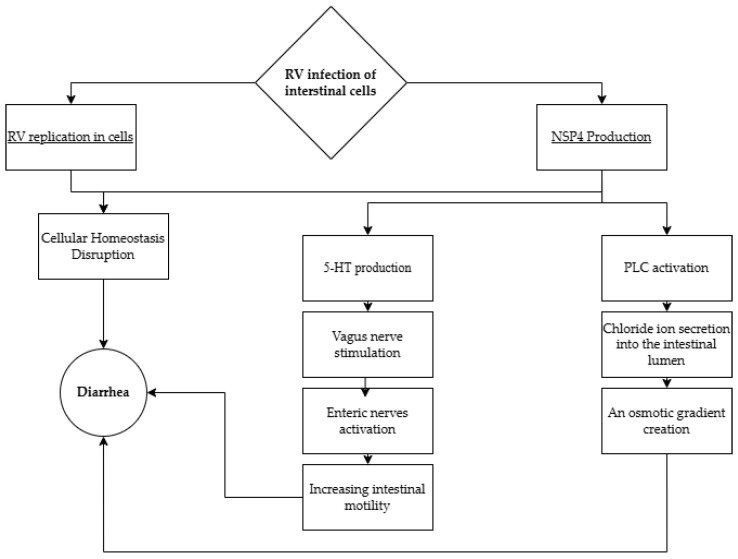
Mechanism of diarrhea caused by RV infection. PLC—phospholipase C; 5-HT—5-hydroxytryptamine, serotonin [29,32,35,37,39].

Diarrhea and the resulting dehydration can be assessed using various scoring systems, including the Vesikari scale, the Clark scale for severity of AGE, the clinical de-hydration scale (CDS), and the Dehydration: Assessing Kids Accurately (DHAKA) score. Several alternative methods for assessing hydration status have also been explored, although these typically require specialized equipment. These include ultrasound-based measurements of the inferior vena cava (IVC) diameter, IVC-to-aorta diameter ratio, aorta-to-IVC ratio, and IVC inspiratory collapse. However, as noted by the ESPGHAN/ESPID Working Group, while these scales can be helpful, there is currently no universally accepted standard for assessment, which will be supported below [40].

The Vesikari scale [41] and the Clark scale [17] are both widely used not only to assess the severity of AGE but also to evaluate the efficacy of RV vaccines. Table 2 provides a comparison of these scales, outlining the parameters assessed and the point allocations for each component [41,42,43].

In Table 3, a comparison of diarrhea severity classification based on the number of points obtained in both scales is presented [43].

A comparative study conducted in London, involving 200 children with RV-induced AGE, found that 57% were classified as severe according to the Vesikari scale, while only 1.5% were classified as severe according to the Clark scale (*p* < 0.001) [43]. Furthermore, 24% of cases classified as mild according to the Clark scale were deemed severe by the Vesikari scale. This discrepancy may be attributed to the absence of a dehydration component in the Clark scale, which reduces its sensitivity for identifying severe cases in comparison to the Vesikari scale. Conversely, the Vesikari scale may not be suitable for comparing the severity of RV disease across different settings, as one of its parameters—treatment status—is influenced by local medical practices and healthcare infrastructure [43,45]. A study conducted in India also demonstrated that the Clark scale tends to underestimate cases that appear clinically severe and require intravenous rehydration [43,46]. Consequently, the Vesikari scale is currently regarded as the most effective tool for identifying endpoints in clinical trials assessing severe RV gastroenteritis in the context of RV vaccine efficacy [39,43,47,48]. Attempts to modify the classification thresholds in these scales have not led to a strong correlation between them [43].

In terms of dehydration assessment, the CDS is utilized, wherein a score of 0 indicates no dehydration, a score of 1–4 represents mild dehydration, and 5–8 indicates moderate to severe dehydration [Table 4] [40,49]. Based on this assessment, clinical decisions regarding treatment, potential hospitalization, or outpatient care can be made. Indications for hospitalization due to AGE include significant dehydration (exceeding 9% of body weight), neurological disturbances (e.g., stupor, seizures), persistent or bilious vomiting, failure of oral rehydration therapy, suspected surgical conditions, and lack of adequate conditions for safe outpatient monitoring and home management [40,50]. The DHAKA Dehydration Score includes four clinical signs: general appearance, presence of tears, respiratory pattern, and skin turgor (skin pinch) [49].

Due to the absence of an ideal scale for evaluating the severity of diarrhea and dehydration, particularly beyond RV etiology—N. Chilyabanyama et al. proposed the Centre for Infectious Disease Research in Zambia (CIDRZ) Scale [51]. The components of this tool are presented in Table 5. The CIDRZ severity assessment tool, which integrates clinical features from existing major scoring systems, demonstrated greater precision and reproducibility compared to the Vesikari (AUC: 0.26) and Clark scales (AUC: 0.18) in predicting hospitalization or the need for intravenous rehydration in children under five years of age (AUC: 0.59) and showed similar performance to the DHAKA Score (AUC: 0.59) [51].

Despite ongoing efforts, an ideal, universally accepted scale for accurately assessing the severity of diarrheal illness and dehydration across different clinical and geographic contexts has yet to be established.

Severe vomiting accompanied by diarrhea is the main factor of dehydration and electrolyte imbalance, which can lead to hospitalization, shock, and even death if not treated properly. A high body temperature (over 38.9 °C) may be associated with dehydration [52].

Patients with AGE have a higher risk of developing kidney stones and acute kidney injury (AKI) via the dehydration mechanism, which results in kidney hypoperfusion. Special attention should be given to children with preexisting kidney disease. A case was reported of a boy with hereditary hypouricemia who developed AKI after AGE [53].

### 4.2. RV Infection and Central Nervous System (CNS)

It was previously believed that RV only infects mature, differentiated enterocytes of the small intestine, due to the prominent symptom of diarrhea in RV infection. However, detection of RV antigens and/or RNA in blood, immune cells, and various organs has provided evidence for systemic dissemination, potentially via hematogenous routes. After oral infection of mice with murine RV, RV-specific proteins were detected in macrophages and B cells in the gut-associated lymphoid tissue (GALT) [54]. This suggests that the lymphatic system may serve as an additional route for virus dissemination beyond the intestines [55].

Antigenemia and viremia are frequently detected in RV A-infected children even when there is no diarrhea. These patients later exhibit increased severity in terms of fever, vomiting, or convulsions [55,56]. In addition to antigenemia and viremia during RV infection, other clinical symptoms are associated with infection in humans, including those within the CNS.

The enterotoxin NSP4 is considered one of the mediators causing neurological complications [57]. Viral proteins have been identified in axons and dendrites, and direct RV infection of neuronal cells has been demonstrated in vitro [58]. The hypothesis regarding the ability of RV to replicate in the CNS and thus influence neurotransmitter dysregulation is controversial, as RV RNA is not always detected in cerebrospinal fluid.

Another hypothesis suggests that circulating mediators, including prostaglandins, cytokines, reactive oxygen species, RV RNA, or NSP4, may act as secondary messengers and indirectly induce effects in the central nervous system. Increased concentrations of excitatory amino acids in cerebrospinal fluid may also cause neurological disturbances and be related to the severity of infection [59].

The blood-brain barrier consists of a highly selective semipermeable border of endothelial cells [60], but some areas of the CNS, such as the area postrema, subfornical organ, pineal gland, and median eminence of the hypothalamus, collectively known as the circumventricular organs, have fenestrated capillaries allowing some vascular permeability. These areas may serve as potential entry points for blood-borne RV to access the brain [54]. Several mechanisms aim to explain CNS complications caused by RV infection, but many aspects remain to be studied.

Although RV has been identified in the CNS, its occurrence is considered rare [4]. Neurological symptoms are reported in approximately 2–6%, but the actual value is unknown [61]. In 1978, Salmi et al. first reported two cases of children with RV-induced AGE, one of whom developed encephalitis with a slow recovery process and the other, fatal Reye’s syndrome [61,62].

Currently described CNS complications associated with RV infection, to be presented in Table 6, can be classified based on features: (1) benign convulsions with mild gastroenteritis; (2) acute encephalopathies/encephalitis; (3) acute cerebellitis; and (4) neonatal RV-associated leukoencephalopathy [63,64,65,66,67,68,69]. Additional reports have described associations with Guillain–Barré syndrome and hemorrhagic shock in pediatric RV cases [61].

### 4.3. RV, Autoimmunity, and Biliary Atresia

The role of RV in inducing autoimmunity has been demonstrated in numerous studies, both in human and animal models, suggesting its potential involvement in the pathogenesis of autoimmune diseases. Epidemiological and immunological evidence indicates a link between RV infection and two autoimmune diseases: celiac disease (CD) and type 1 diabetes mellitus (T1DM) [70]. RVs have also been observed to migrate to the liver and replicate within the epithelial cells of the biliary tree [71]. One of the proposed mechanisms underlying this association is molecular mimicry, whereby RV antigens resemble host autoantigens and may trigger an autoimmune response [72].

Among the viral proteins implicated, the VP7 protein has been shown to share molecular similarities with several key human autoantigens. These include IA-2 and GAD65, both of which are targets in T1DM [72,73,74,75], as well as transglutaminase, the main autoantigen involved in CD [76,77]. Another viral protein, VP4, has been associated with molecular mimicry of α-enolase, a proposed autoantigen in biliary atresia [78]. These findings support the hypothesis that RV infection may act as a trigger for autoimmunity through cross-reactive immune responses, although further investigation is necessary to confirm the causality and underlying mechanisms [72].

Cohort studies in the USA [79], encompassing 1,474,535 children, led by Rogers et al. from 2001–2017, suggested that a complete routine vaccination schedule against RV A reduces the incidence of type 1 diabetes in children. Other less extensive cohort studies, involving 880,629 children, present an association between RV A vaccination and reduced incidence of type 1 diabetes [80]. In early research, RV infections were identified as possible triggers for T1DM, as suggested by sequence homology between viral peptides and those of pancreatic islet cell proteins in affected individuals. Additionally, RV infection increased the risk of T1DM in NOD mice. However, studies on the relationship between RV infections and the risk of developing T1DM in humans have yielded mixed results, suggesting that additional contributing factors, such as age and diet, may play significant roles [70].

Antibodies against the RV VP7 protein modulate the expression of genes involved in apoptosis and inflammation of the intestinal epithelium, leading to changes in epithelial barrier integrity, typical of celiac disease. A prospective study conducted on nearly 2000 children with a genetic predisposition to CD indicated that a high frequency of RV infections might increase the risk of developing CD [81]. However, there is currently insufficient evidence to confirm a direct impact of RV infections on the development of this disease [82].

To induce biliary atresia, a significantly high RV dose, approximately 10,000 times higher than the usual infectious dose, is necessary. The reason for this high dosage requirement is unclear, but it is likely because a large number of cholangiocytes need to be infected to initiate the immune responses leading to biliary atresia [83]. The conducted study showed that cholangiocytes exhibit a notable resistance to infection, which could explain why this condition is age-dependent, with neonatal cholangiocytes being more susceptible to RV infection than those in older mice [84]. Studies indicate that only certain RV strains are capable of causing biliary atresia, with strain-specific features determining their preference for hepatobiliary cells and thus influencing the disease’s onset. These findings underscore that RV’s ability to infect and replicate within the biliary epithelium is critical for initiating the pathogenic cascade leading to biliary atresia [85]. Additionally, the studies underscore that finding the virus in liver tissue does not always coincide with the development of biliary atresia, as the virus might be replicating in cells other than biliary ones, which are unlikely to lead to the condition [86].

### 4.4. RV and Respiratory Tract Infections (RTIs)

Respiratory tract infections (RTIs), particularly lower respiratory infections, represent the second leading cause of disability-adjusted life-years (DALYs) in the 0–9-year age group [87]. Studies suggest a potential connection between RTIs and RV, which is mainly associated with gastrointestinal tract diseases [88]. In the study conducted by G. T. Zhaori et al., RV antigens were detected in tracheal aspirates from children diagnosed with clinical pneumonia, suggesting a potential association between RV infection and respiratory tract involvement [89]. RV was also detected in the oropharynx squamous and goblet cells and epithelial cells from the respiratory tract in infants hospitalized with respiratory symptoms in the study led by B. J. Zheng, et al. Similar viral RNA was also detected in stool samples of children with diarrhea in this study [90]. C. D. Brandt et al. highlighted the possibility of upper respiratory tract infection caused by RV occurring simultaneously with gastroenteric infections caused by different types of viruses, such as adenoviruses, with a co-infection rate of 24.3% [91]. Further research is warranted to determine the extent of RV involvement in respiratory tract pathology, particularly through studies that investigate the virus’s ability to bind and enter respiratory epithelial cells via specific surface receptors such as integrins or sialic acid-linked glycans [92]. Studies should also explore the replication dynamics of RV in airway tissues, the extent of epithelial barrier disruption, and the induction of local cytokine and chemokine responses, including IL-6, IL-8, and interferons, caused by RV. These investigations would allow clarification of whether RV contributes directly to respiratory disease pathogenesis or exacerbates symptoms through immunomodulatory effects. Moreover, these studies may contribute to improving antiviral therapeutic strategies targeting RV [93].

## 5. AGE RV^+^ Progression Across Various Age Groups

The progression of RV infection starts with an incubation period of 16–18 h, followed by early symptoms. In addition to vomiting and fever, other symptoms may include loss of appetite and dehydration, characterized by dry mouth, decreased urination, and crying with few or no tears [94]. It then details the main symptoms, including diarrhea lasting 3–8 days with mucus in the stool, vomiting, diarrhea, and fever. RV infections can manifest with headaches, nausea, muscle aches, abdominal discomfort, loss of appetite, and fatigue. These symptoms, collectively known as sickness symptoms, can appear early in the course of an RV infection, even before the onset of diarrhea [95].

### 5.1. Disease Progression in Children Under 5 Years of Age

The peak incidence of RV infections in children under 5 years of age in developed countries, according to European studies, occurs in the second year of life, which is also reflected in studies from Spain [96]. In developing countries, this peak occurs between 6 and 12 months due to factors such as inadequate water supply, poor sanitation and hygiene conditions, and low vaccination rates, which significantly contribute to higher transmission intensity of RV and an earlier peak incidence compared to their counterparts in high-resource settings [97].

A meta-analysis of studies on RV diarrhea in children under 5 years in Ethiopia showed that the likelihood of diarrhea was three times higher in children whose mothers did not wash their hands at critical moments and also three times higher if the mothers had diarrhea in the previous two weeks. Children in households not using treated drinking water had twice the risk of diarrhea compared to those from households using treated water. Improving water quality has proven to be an effective strategy for preventing diarrheal diseases [98].

The highest susceptibility in this age group generally corresponds to the decline in immune antibodies and other factors acquired from the mother, which usually occurs after 5 months of age. It should be noted, however, that susceptibility to RV disease persists throughout life, with the most severe cases occurring in infants during their first infection [96].

Children with RV-induced AGE under 3 years of age are more likely to present to the emergency department than children with RV-negative AGE (41.5% vs. 25.0%, *p* < 0.001) [99]. Among enteropathogenic viruses causing AGE in Africa, RV accounts for 31.0% of all gastroenteritis cases in hospitalized and outpatient patients, with an estimated prevalence of 22% to 37% across all geographic areas in Africa [100].

Studies in Qatar indicate that most children under 5 years of age hospitalized for RV-induced AGE had severe symptoms of diarrhea (100%), dehydration (96.5%), vomiting (97%), and fever (54%) [97]. The maximum number of vomiting episodes within 24 h and their duration were dramatically higher in RV^+^ patients than RV^−^ according to studies in Spain, but this is not confirmed by studies conducted in Qatar [62,68]. The prevalence of watery stools and dehydration was significantly higher in RV^+^ children than in RV^−^ children [101]. RV^+^ children cried more (73.5% vs. 51.4%, *p* < 0.001), were more irritable (76.5% vs. 59.8%, *p* < 0.001), and were more tired than usual (77.5% vs. 54.2%, *p* < 0.001) compared to RV-negative children [99].

The Global Enteric Multicenter Study (GEMS) identified 12 pathogens with the highest prevalence: adenovirus 40/41, astrovirus, *Campylobacter jejuni* or *Campylobacter coli*, *Cryptosporidium* spp., norovirus GII, RV, *Salmonella* spp., sapovirus, *Shigella* spp., heat-stable toxin-producing enterotoxigenic Escherichia coli (ST-ETEC), typical enteropathogenic *E. coli* (EPEC), and *Vibrio cholerae* [102]. Among these pathogens, according to global analysis, RV is the leading cause of diarrheal deaths in children under 5 years of age, although its share may have decreased from 26.5% in 2000 to 24.4% in 2021 [103].

### 5.2. Disease Progression in Children over 5 Years of Age

A systematic review and meta-analysis of the published literature, encompassing 66 studies from 32 countries in 2021 [104], according to the authors, is the first such analysis of RV prevalence among older children and adults experiencing diarrhea globally. Some studies included in the meta-analysis found higher RV prevalence among older children and/or older adults compared to younger adults. However, not all studies demonstrated this phenomenon [104]. The pooled prevalence of RV among individuals with diarrhea aged 5 years and older was 7.6%, but prevalence varied widely between studies [104].

Given that natural RV infection protects against subsequent infections, children over 5 years old are rarely affected by RV infection. However, a study conducted at Nihon University Hospital in Chiyoda-ku, Tokyo, which included children under 15 years of age hospitalized with AGE in this institution over five years (from January 2015 to December 2019), indicates an increase in the prevalence of infections among children older than 5 years in this area [105]. Despite the significant decline in the incidence of severe RV infections due to the introduction of RV vaccines, it should not be assumed that RV infections cannot occur in older children and that these infections are exclusively infant-specific. One possible reason for RV infections observed in older children is that they may not have been naturally exposed to RV in infancy, likely due to the overall reduction in community-level circulation of the virus following widespread vaccination efforts in Japan. Consequently, their immune systems might lack sufficient natural immunity, leaving them susceptible at a later age. Another potential explanation is the circulation of RV genotypes that differ antigenically from those included in the vaccine or from previously encountered strains. This antigenic variation could allow infection to occur even in previously vaccinated individuals, although in this particular study, none of the patients had a confirmed history of natural RV infection [105]. It should also be noted that the Vesikari score [41], describing the severity of clinical symptoms in these children, was significantly lower in vaccinated patients compared to unvaccinated ones.

RV infections should not be treated as infections exclusive to children under 5 years of age. The high heterogeneity of the review results [105] indicates the need to improve surveillance systems for RV infections in older children and adults, especially in the African region, and to consider changes resulting from the introduction of RV vaccination into global immunization programs.

## 6. Vaccination and Its Efficacy in Preventing RV Infections

### 6.1. The Types of Available Vaccines with Dosing Schedules

The types of available vaccines, both WHO-approved and nationally licensed, along with their composition, are presented in Table 7 [106,107,108,109,110,111].

The WHO recommends administering the first dose of the vaccine as soon as possible after 6 weeks of life, along with the DTP vaccination. Table 8 [112] presents the dosing schedules depending on the type of preparation.

### 6.2. Adverse Effects Associated with Vaccine Administration

Regarding adverse effects associated with vaccine administration, according to the Rotarix Summary of Product Characteristics [107], irritability and loss of appetite are very common; diarrhea, flatulence, abdominal pain, vomiting, fever, and weakness are common; constipation is uncommon; and hoarseness or muscle cramps are rare. As reported by the manufacturer of Rotateq [106], diarrhea, vomiting, fever, and upper respiratory tract infection are widespread side effects. Intussusception has also been associated with RV vaccines. This discovery led to the withdrawal of the first-generation tetravalent RV vaccine (RRV-TV) in 1999 due to its association with intussusception. However, a review of studies in Chongqing did not show a significant statistical correlation between vaccine administration and the occurrence of intussusception, regardless of the brand or type of vaccine (excluding RRV-TV, which was not analyzed) [113].

### 6.3. Contraindications to Vaccine Administration

Contraindications to vaccine administration include severe hypersensitivity to any of its components, severe immunodeficiency, including severe combined immunodeficiency, and past occurrence of intussusception in the child to be vaccinated. Acute diarrhea or acute febrile illness is a relative contraindication to deferring vaccination. Both the Rotarix and RotaTeq vaccines have demonstrated good tolerance in preterm infants, HIV-infected infants, and those exposed to HIV but not infected. The Global Advisory Committee on Vaccine Safety (GACVS) has emphasized that the benefits of vaccination outweigh the minimal risk of intussusception [112].

### 6.4. Effectiveness

A meta-analysis encompassing 34 African countries that introduced RV vaccines into their national immunization programs, using both the monovalent (Rotarix^®^, RV1) and pentavalent (RotaTeq^®^, RV5) vaccines, confirms a significant reduction in the proportion of RV-positive cases from 42% (95% CI: 38–46) in the pre-vaccine period to 21% (95% CI: 17–25) in the post-vaccine period. These findings are consistent with observations in the Caribbean and Latin American regions, which demonstrated that RV vaccines effectively protect children against hospitalizations due to diarrhea [114,115].

The induction of IgA antibodies is regarded as a crucial predictor of protection against RV infection. However, serum IgA titers are more strongly associated with protection against RV diarrhea in high-income countries than in low- and middle-income countries. Therefore, it was deduced that vaccine efficacy is higher in high-income countries than in middle- and low-income countries. In some trials, type-specific neutralizing antibody titers have also been reported [116]. Higher vaccine efficacy has also been noted in countries with low mortality rates compared to those with high mortality rates, the cause of which remains unknown [3]. In medium- and high-mortality settings, a decline in vaccine efficacy from the first to the second year of life has been suggested [3,117,118,119], which was not observed in two age groups in the low-mortality-rate cohort [120]. A meta-analysis in Sub-Saharan African countries similarly indicates that vaccines provided significant protection against hospitalizations due to RV diarrhea in children under 12 months compared to those over 12 months of age [114].

There are multiple potential reasons for the disparity between the effectiveness of vaccination in middle- and low-income countries. Factors such as nutritional status, composition of the gut microbiota, and variations in immune responses significantly contribute to the disparity in vaccine efficacy between high- and low-income countries. The composition of gut microbiota, which varies significantly between populations in high- and low-income countries, plays a crucial role in modulating immune responses to live vaccines such as the RV vaccine. The gut microbiota of children in low-income countries tends to exhibit greater diversity and variability over time compared to that of children in high-income countries [121,122,123]. Nutritional factors can significantly affect the functionality of the adaptive immune system, influencing how children respond to vaccines. For instance, vitamin A deficiency can disrupt the trafficking of vaccine-specific CD8+ T cells to the gastrointestinal tract, while zinc deficiency is linked to impaired T cell function and depressed immune responses [124,125].

### 6.5. Vaccination Coverage

Since 2000, mortality due to diarrheal diseases has decreased by 60%, but 497,434 child deaths were estimated to have occurred in 2019. Etiologically, the leading cause of these fatal infections is RV (accounting for 30% of diarrhea-related deaths in children under 5 years of age) [126]. In 2016, RV infection was the cause of infection in 128,500 children under 5 years of age, the majority of whom were in Sub-Saharan Africa. It is estimated that in 2016, vaccinations prevented 28,000 deaths of children under 5 years of age, and their broader application could prevent approximately 20% of deaths [127].

The WHO recommends that RV vaccines be included in all national vaccination programs, with particular emphasis on countries in South and Southeast Asia and Sub-Saharan Africa [112]. The percentage of the population vaccinated against RVs based on WHO data [Figure 3] shows a continuous increase in the percentage of the global RV vaccination coverage [128].

### 6.6. Vaccination Dropout

The dropout rate, defined as the difference between the percentage of children receiving the first and final doses, remains a significant issue, particularly in Sub-Saharan Africa [128]. According to a study conducted in Ethiopia, this rate was 1.9% nationwide [129]. Studies conducted in Kenya indicate a dropout rate of 9.1% in the Kiambu sub-district and 17.1% in the Mbita sub-district [130]. An analysis of 21 Sub-Saharan African countries (including Ethiopia, Angola, Burundi, Cameroon, Kenya, Senegal, Gambia, Malawi, Liberia, Mali, Sierra Leone, Rwanda, Zimbabwe, South Africa, Tanzania, Uganda, and Zambia) from 2015–2022 showed that 10.77% of children who received the first dose of the RV vaccine did not receive the second dose (95% CI: 10.55–11.00%). Dropout rates varied from 2.77% in Rwanda to 37.67% in Uganda. The significant dropout rates in these countries, combined with WHO recommendations, highlight the need to analyze the causes and find solutions to increase vaccination coverage in Sub-Saharan Africa [128].

Statistical analyses have shown that factors such as young maternal age (relying on decisions or approval from older family members), home births (leading to limited access to vaccination schedules), lack of postnatal and antenatal care, limited access to healthcare facilities, and maternal unemployment are associated with a higher likelihood of skipping the second dose of the RV vaccine [128].

### 6.7. RV Vaccines and Dilemmas: Vaccinating Immunocompromised Children, Premature Infants, and High-Medical-Risk Infants

Contraindications to vaccine administration include severe hypersensitivity to any of its components, severe immunodeficiency including severe combined immunodeficiency, and past occurrence of intussusception in the child to be vaccinated.

The RV vaccine (RV) is effective in preventing severe RV-induced diarrhea but is contraindicated in infants with severe immunodeficiencies, including severe combined immunodeficiency (SCID). This is due to severe adverse effects such as severe diarrhea, vomiting, fever, or failure to gain weight in vaccinated infants with these health problems [131,132,133].

In HIV-infected infants or those exposed but uninfected (HEU), administration of the RV vaccine did not increase the incidence of adverse events, as confirmed by a study on the safety of RV vaccines in HIV-infected infants and HEU children conducted in four African countries [131,134]. A study in Kenya involving 1308 infants randomly assigned to receive RV or placebo, with 37 children infected with HIV and 177 exposed but uninfected, showed no significant difference in the incidence of severe and mild adverse events between these groups [131,135].

The safety of administering the RV vaccine to infants born to mothers with immunosuppression due to the increasing use of biologics during pregnancy for treating autoimmune diseases is also gaining importance. Data from the U.S. PIANO registry indicate certain adverse events following fetal exposure to infliximab. Among 43 RV-vaccinated infants, 6 (15%) had a fever, and 1 (3%) had diarrhea [136]. In other studies [137,138,139], no adverse events were reported in these mothers’ infants post-RV vaccination. Due to limited research, it is recommended to avoid administering the RV vaccine to infants exposed to immunosuppressive drugs during the fetal period until at least 6–12 months of age, per guidelines from the UK and the US [140,141].

Preterm infants who remain hospitalized beyond 2 months of age often do not receive the RV vaccine due to concerns about severe disease caused by the attenuated vaccine virus strain, the possibility of virus transmission to other infants in the neonatal intensive care unit (NICU), and reduced vaccine efficacy. However, preterm infants are more susceptible to severe RV infection, and previous studies have shown higher hospitalization rates and more severe gastrointestinal symptoms, including bowel distension, abdominal bloating, and mucoid stools [142]. Findings on RV vaccine safety and tolerability in preterm infants from previous studies indicate that rates of adverse events (AEs) are similar to those observed among term infants [143]. Different countries have different guidelines for RV vaccinations. The UK and Australia promote vaccinating infants during their NICU stay according to standard schedules. In Canada, vaccinations are considered on a case-by-case basis for each facility. In the US, it is recommended that RV vaccines not be administered until discharge from the hospital, although many hospitals still provide RV vaccines to hospitalized infants [131].

In infants at high medical risk requiring long-term care (including preterm infants, those with low birth weight, or congenital disorders), a single dose of the human RV vaccine provides limited protection against severe RV AGE. A higher incidence of serious adverse events (SAEs) related to the vaccine was observed in comparison to healthy infants. The human RV vaccine was generally well-tolerated, but its co-administration with vaccinations of their national immunization program increased the risk of gastrointestinal adverse events by 8%. These findings differ from those for healthy infants and underscore the need for vaccine research in specific vulnerable populations [143].

### 6.8. Perspectives on Next-Generation RV Vaccines

As current vaccines have significantly reduced the global burden of severe RV gastro-enteritis, there is growing attention towards developing next-generation vaccines aimed at addressing existing limitations, such as strain-specific efficacy, heat stability, and cost-effectiveness. Highlighting these novel approaches is essential to understanding future directions in RV prevention and improving global vaccine coverage and effectiveness. Table 9 presents an overview of next-generation RV vaccines, detailing their composition, developer, and clinical trial phase [144,145,146,147,148,149,150,151,152,153,154,155,156,157,158,159,160,161].

## 7. Concluding Remarks and Perspective

This review provides a comprehensive and up-to-date analysis of RV infections, combining detailed insights into the virus’s pathophysiology, clinical manifestations, and vaccination strategies. A key contribution of the work is the thorough examination of both local and systemic effects of RV, including less frequently discussed complications such as central nervous system involvement, autoimmune responses (T1DM, CD), and possible links with biliary atresia and respiratory tract infections. By integrating clinical scales such as Vesikari, Clark, and CDS/DHAKA, the review also offers a practical framework for assessing disease severity.

Furthermore, the review evaluates the effectiveness and limitations of current vaccines, highlighting challenges in immunization coverage, particularly in Sub-Saharan Africa and in vulnerable groups such as preterm and immunocompromised infants. The comparison of first-generation and next-generation vaccine candidates presents a valuable outlook on future directions in RV prevention.

Importantly, the review brings attention to the potential for interspecies transmission of RVs, referencing recent studies and drawing parallels with current concerns about SARS-CoV-2. Additionally, growing interest in the zoonotic transmission of SARS-CoV-2 draws attention to the potential role of animals not only in spreading this virus but also in the transmission of RVs, underscoring the importance of further investigations into interspecies transmission pathways and reservoirs of infection [162].

By integrating clinical, immunological, and epidemiological data, this work supports improved disease recognition, enhances understanding of RV beyond gastrointestinal involvement, and lays the groundwork for more targeted prevention and therapeutic approaches.

## Figures and Tables

**Figure 1 pathogens-14-00480-f001:**
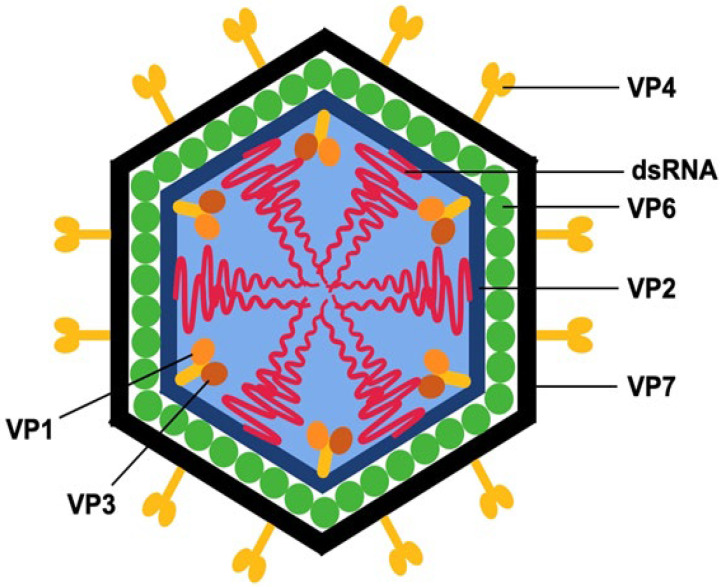
Localization of RV proteins within the viral particle [10,11].

**Figure 3 pathogens-14-00480-f003:**
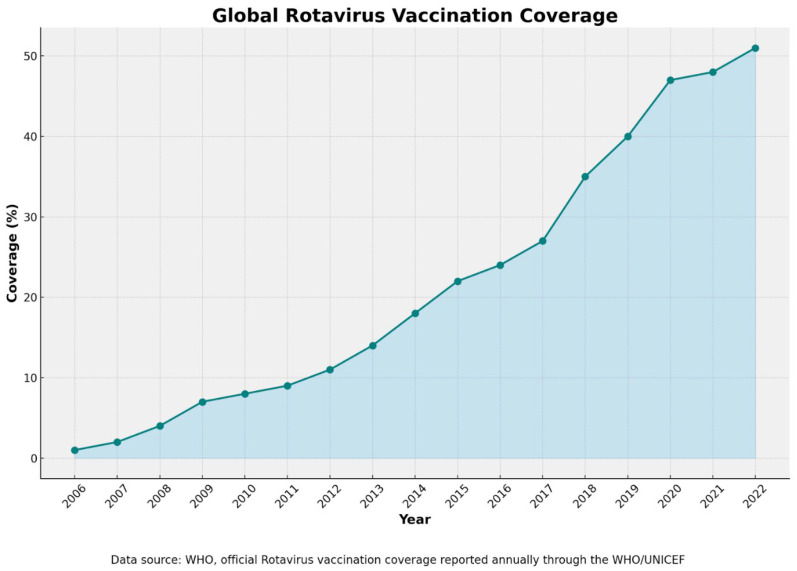
The percentage of the population vaccinated against RVs based on WHO data (2006–2022) [128].

**Table 1 pathogens-14-00480-t001:** Functional characterization of RV structural and non-structural protein [10,11,12,13,14,15,16,17].

Name of Protein	Role of the Protein
**Structural proteins (VP)**	
VP4, VP7	Mediate viral attachment to host cell surface receptors and facilitate membrane penetration during entry. VP4 contributes to viral antigenic diversity, with 58 recognized P genotypes [12].
VP1	Functions as the RNA-dependent RNA polymerase, catalyzing the synthesis of viral RNA [12].
VP2	Serves as the inner capsid scaffold and essential cofactor for VP1, enabling initiation of double-stranded RNA genome replication [11].
VP3	Guanylyl-methyltransferase, capping enzyme [10]. RV proteins NSP1 and VP3 act to suppress interferon (IFN) expression by inducing degradation of transcription factors and other host elements essential for effective innate immune responses [11].
VP6	Major structural protein [13]. Involved in RV diversity—42 G-types and species classification (RVA–RVJ).
**Non-structural proteins (NSP)**	
NSP1	Interferon antagonist [14].
NSP2	Involved in RV particle assembly [14,15].
NSP3	Stimulates translation of both capped and uncapped viral mRNA [16].
NSP4	Involved in RV particle assembly, enterotoxin [14]
NSP5	Involved in RV particle assembly [14,17].
NSP6	Expressed from an alternative reading frame of the NSP5 gene in some Group A RV strains [15].

**Table 2 pathogens-14-00480-t002:** A comparison of the Vesikari and Clark Scales, detailing the components and the respective point allocations for each scale [44].

Symptoms	Vesikari Scale			Clark Scale		
	1	2	3	1	2	3
Number of stools/days	1–3	4–5	≥6	2–4	5–7	≥8
Duration of diarrhea (days)	1–4	5	≥6	1–4	5–7	≥8
Number of emesis events/day	1	2–4	≥5	1–3	4–6	≥7
Duration of emesis (days)	1	2	≥3	2	3–5	≥6
Rectal temperature (°C)	37.1–38.4	38.5–38.9	≥39	38.1–38.2	38.3–38.7	≥38.8
Temperature duration (days)	–	–	–	1–2	3–4	≥5
Dehydration	–	1–5%	≥6%	–	–	–
Behavioral symptoms	–	–	–	Irritable/less playful	Lethargic/listless	Seizures
Duration of behavioral symptoms (days)	–	–	–	1–2	3–4	≥5
Treatment	Rehydration	Hospitalization	–	–	–	–

**Table 3 pathogens-14-00480-t003:** Comparison of Vesikari and Clark scale severity based on points obtained in both scales [43].

The Vesikari Scale		The Clark Scale	
Number of Points Obtained in Scale	Severity	Number of Points Obtained in Scale	Severity
<11	Non-severe	0–8	Mild
-	-	9–16	Moderate
≥11	Severe	>16	Severe

**Table 4 pathogens-14-00480-t004:** Table of CDS and DHAKA dehydration score detailing the components and the respective point allocations for both scales [40,49].

Clinical Dehydration Scale (CDS)				DHAKA Scale		
Characteristics/Points	0	1	2	0	2	4
General appearance	Normal	Thirsty, restless, or lethargic but irritable when touched	Cold, drowsy, limp, sweaty, or comatose	Normal	Restless, irritable	Lethargic, unconscious
Eyes	Normal	Slightly sunken	Extremely sunken	-	-	-
Mucous membrane (tondue)	Moist	Sticky		-	-	-
Tears	Normal	Decreased tears	Absent tears	Normal	Decreased	Absent
Skin pinch	-	-	-	Normal	Slow	Very slow
Respirations	-	-	-	Normal	Deep	-

CDS: 0—no dehydration, 1–4—mild dehydration, 5–8—moderate to severe dehydration. DHAKA Dehydration Score Categories: 0–1—no dehydration, 2–3—some dehydration, ≥4—severe dehydration.

**Table 5 pathogens-14-00480-t005:** Centre for Infectious Disease Research in Zambia (CIDRZ) Scale [51].

Clinical Parameter	Assessment Criteria	Points
Vomiting frequency	2–3 episodes/day	1
	4–5 episodes/day	2
	≥6 episodes/day	3
Behavioral status	Restless/irritable	2
	Lethargic	3
Skin pinch test	Normal (instant recoil)	0
	Slow recoil (>2 s)	2
	Very slow	3
Tears	Present	0
	Absent	2
Respirations	Normal	0
	Deep	2

CIDRZ: ≥5—Cut-off from severe diarrhea.

**Table 6 pathogens-14-00480-t006:** Types of CNS complications associated with RV infection [63,64,65,66,67,68,69].

Types of CNS Complications	Clinical Features
Benign convulsions with mild gastroenteritis	After the onset of gastroenteritis, febrile or afebrile seizures. Seizures may precede the onset of diarrhea by 12 to 24 h or coincide with its initiation. Clinically, seizures predominantly manifest as generalized tonic-clonic episodes [63,64].
Acute cerebellitis	After 1–3 days, the onset of symptoms of gastroenteritis decreased consciousness and subsequent mutism (may last for up to 20 days), slow speech, dysarthria, hypotonia, ataxia, tremors, nystagmus, and dysmetria [65].
Neonatal RV-associated leukoencephalopathy	At around the 5th day after birth, repetitive or clustered focal or multifocal clonic seizures [66].
**Acute Encephalopathies/Encephalitis**	
Mild encephalopathy with a reversible splenial lesion (MERS)	Prodromal symptoms: fever, cough, vomiting, and diarrhea. Decreased consciousness, seizures, and delirious behavior [67,68,69].
Acute encephalopathy with biphasic seizures and late reduced diffusion (AESD)	During days 1–2, a prolonged febrile seizure occurs, which is followed by a cluster of complex partial seizures with impaired consciousness between days 3 and 7. In addition, 20–30% of patients experience an interval during which consciousness remains normal and no neurological symptoms are evident [69].
Acute necrotizing encephalopathy (ANE)	Rapid progression of altered consciousness and seizures. Vomiting and diarrhea are common [69].

**Table 7 pathogens-14-00480-t007:** The types of available vaccines [106,107,108,109,110,111].

Vaccination Name	Basic Concept	National License
RotaTeq [106]	Live-attenuated, human-bovine reassortant vaccine: 5 reassortant strains in one vaccine containing human G1, G2, G3, G4 (VP7), and P (VP4) inserted into the bovine G6P.	WHO-prequalified vaccine
Rotarix [107]	Human, live-attenuated G1P RV vaccine.	WHO-prequalified vaccine
Rotavac [108]	Live-attenuated, naturally reassorted human-bovine single strain G9P vaccine, containing one bovine RV gene P and 10 human RV genes, including G9.	WHO-prequalified vaccine
ROTASIIL [109]	Live-attenuated, human-bovine reassortant vaccine: 5 reassortant strains in one vaccine containing human G1, G2, G3, G4, and G9 (VP7) inserted into the bovine G6P UK strain.	WHO-prequalified vaccine
Rotavin-M1 [110]	Human, live-attenuated G1P RV vaccine.	National license granted by Vietnam in 2012.
LLR (Lanzhou lamb RV vaccine) [111]	Lamb, live-attenuated G10P RV vaccine.	National license granted by China in 2000.

**Table 8 pathogens-14-00480-t008:** Vaccination dosing schedules depending on the type of preparation [112].

Vaccination Name	Vaccination Dosing Schedules
RotaTeq	Beginning at 6–12 weeks of age, 3 oral doses given 4–10 weeks apart; the series should be completed by the age of 32 weeks.
Rotarix	Beginning at 6 weeks of age; 2 oral doses given 4 weeks apart; the series should be given before 16 weeks of age but must be completed by the age of 24 weeks.
Rotavac	Beginning at 6 weeks of age; 3 oral doses given 4 weeks apart; the series should be completed before the age of 8 months.
ROTASIIL	Beginning at 6 weeks of age, 3 oral doses given 4 weeks apart; the series should be completed during the first year of life.
Rotavin-M1	2 oral doses administered 2 months apart; the first dose must be administered at 6–12 weeks of age.
LLR (Lanzhou lamb RV vaccine)	3 oral doses: 1 dose per year for 3 consecutive years in children aged 2–36 months.

**Table 9 pathogens-14-00480-t009:** Overview of next-generation RV vaccines—composition, developer, and clinical trial phase [144,145,146,147,148,149,150,151,152,153,154,155,156,157,158,159,160,161].

Vaccination Name	Basic Concept	Developer	Development Phase
RV3-BB	Human neonatal G3P[6] RV vaccine.	Murdoch Children’s Research Institute/Biofarma	A Phase II Dose-Ranging Study of Oral RV3-BB RV Vaccine ended [144].
Tetravalent UK-BRV	Live-attenuated, human-bovine reassortant vaccine: 4 reassortant strains, including human G1–4, inserted into the bovine G6P[5] UK strain.	Shantha Biotechnics	A phase I/II development—development abandoned [145,146].
Hexavalent UK-BRV	Live-attenuated, human-bovine reassortant vaccine: 6 reassortant strains, including human G1–4, G8, and G9, inserted into the bovine G6P[5] UK strain.	Developer: Wuhan Institute of Biological Products, China. Development.	A Phase III clinical trial for efficacy and safety in China ended. No further clinical trials have been conducted for this candidate vaccine in Brazil, where the inclusion of Rotarix in the national immunization program has significantly reduced the RV disease burden [147].
Pentavalent UK-BRV	Live-attenuated, human-bovine reassortant vaccine: 5 reassortant strains, including human G1–4 and G9, inserted into the bovine G6P[5] UK strain.	Developer: Instituto Butantan, Brazil. Development.	Phase I clinical trial for the efficacy and safety in Brazil ended [148].
G1P[8], inactivated	Inactivated human RV vaccine.	Centers for Disease Control and Prevention (CDC), USA.	Preclinical development [149,150,151,152].
P2-VP8-P[8] and P2-VP8-P[4/6/8]	Subunit RV vaccines based on recombinant proteins.	PATH RV Vaccine Program, USA.	A phase III safety, immunogenicity, and efficacy study in infants 6 to 8 weeks of age in Ghana, Malawi, and Zambia—development abandoned. Findings indicated insufficient evidence that the trivalent P2-VP8 offers improved protection against severe RV diarrhea compared to currently licensed oral vaccines [153].
MBP::VP6 and pCWA:VP6	Subunit RV vaccines based on recombinant proteins.	Cincinnati Children’s Hospital, USA; Laboratoria de Immunologia y Virologia (LIV), Argentina.	Preclinical development [153,154,155,156].
VP2/6/7 and VP2/4/6/7VLPs	Subunit RV vaccines based on virus-like particles.	Baylor College of Medicine, USA.	Preclinical development [157,158].
VP6 GI.3/GII.4 RV-NoV VLPs	Subunit RV vaccines based on virus-like particles.	University of Tampere School of Medicine, Finland.	Preclinical development [159,160,161].

## Data Availability

No new data were created or analyzed in this study.

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
