# Peer review of "Rotavirus Infections: Pathophysiology, Symptoms, and Vaccination"

_pathogens, 2025, doi:10.3390/pathogens14050480_

Round 1
Reviewer 1 Report
Comments and Suggestions for Authors
The scope of the review is too broad, lacking a clear focus that is thoroughly analyzed.
- Using reviews to write another review seems suboptimal. Consider citing original studies. For example, in line 181, when discussing the identification of RV in the CNS, readers would expect a study describing the direct detection of RV in the CNS. However, the reference provided does not meet these expectations.
- In the section on general infections, do the authors believe that all listed clinical symptoms are directly related to the presence of RV in these tissues?
- For most of the data presented, there is a lack of analysis. Simply listing multiple studies without proper analysis limits the potential impact of the review.
- Consider renaming the manuscript, as the review is not solely focused on pediatric patients. Additionally, the sections on RV diagnosis and vaccination seem somewhat out of scope, making the overall focus too broad.
- Subheading 4.1: Several studies are listed, but are they all directly relevant to the title of this subsection?
- In the "General Infections" section, consider adding information on respiratory disorders associated with RV.
- Line 60: Both RVA and RVB are associated with diarrhea in pigs and other animals.
- Since multiple RV species are mentioned, it would be beneficial to specify the species for all cited studies.
- Line 71: The correct term is "murine."
- The "Local Infection" section lacks general background information.
- Lines 145–149: Is this describing systemic infection?
- Consider including a schematic diagram to illustrate the described mechanisms.
Author Response
We would like to express our sincere gratitude for your thorough and insightful review. Your feedback was exceptionally valuable, and we greatly appreciate your constructive suggestions.
In response to your review, we have decided to revise our manuscript by removing the section concerning diagnostics. The current version of our work focuses primarily on the pathophysiology of symptoms, treatment strategies, and rotavirus vaccination.
We have made a concerted effort to eliminate, as much as possible, references to narrative reviews. The revised manuscript now predominantly cites original research articles, cohort studies, case reports, guidelines issued by European scientific societies, meta-analyses, and systematic reviews.
- Using reviews to write another review seems suboptimal. Consider citing original studies. For example, in line 181, when discussing the identification of RV in the CNS, readers would expect a study describing the direct detection of RV in the CNS. However, the reference provided does not meet these expectations.
- We have made a concerted effort to eliminate, as much as possible, references to narrative reviews. The revised manuscript now predominantly cites original research articles, cohort studies, case reports, guidelines issued by European scientific societies, meta-analyses, and systematic reviews.
- Regarding line 181 (prior to revisions), the structure of the table was modified to better align with the data available in the cited references.
- We appreciate your insightful feedback and thank you for your valuable suggestions.
2. In the section on general infections, do the authors believe that all listed clinical symptoms are directly related to the presence of RV in these tissues?
- Thank you for your valuable comments on this matter. We have made adjustments to the terminology, primarily within this section, in order to provide a clearer distinction between local infections, systemic effects, and potential associations of RV with other diseases.
3. For most of the data presented, there is a lack of analysis. Simply listing multiple studies without proper analysis limits the potential impact of the review.
- We have made a concerted effort to eliminate, as much as possible, references to narrative reviews. The revised manuscript now predominantly cites original research articles, cohort studies, case reports, guidelines issued by European scientific societies, meta-analyses.
4. Consider renaming the manuscript, as the review is not solely focused on pediatric patients. Additionally, the sections on RV diagnosis and vaccination seem somewhat out of scope, making the overall focus too broad.
- The modification has been implemented.
5. Subheading 4.1: Several studies are listed, but are they all directly relevant to the title of this subsection?
- We noticed inconsistencies, and parts of the work that were not relevant have been removed. The remaining articles focus on individuals under 5 years of age or include this age group in their analysis.
6. In the "General Infections" section, consider adding information on respiratory disorders associated with RV.
- The modification has been implemented.
7. Line 60: Both RVA and RVB are associated with diarrhea in pigs and other animals.
- The modification has been implemented.
8.Since multiple RV species are mentioned, it would be beneficial to specify the species for all cited studies.
- The modification has been applied throughout the manuscript to the greatest extent possible.
9. Line 71: The correct term is "murine.
- The modification has been implemented.
10. The "Local Infection" section lacks general background information.
- Previously, information about Local Infections was included in the Introduction. We have changed its location for better clarity and understanding of the topic.
11. Lines 145–149: Is this describing systemic infection?
- Dehydration, as a systemic effect, has been moved to the section on Systemic Infections.
12. Consider including a schematic diagram to illustrate the described mechanisms.
- The modification has been implemented.
- We create Figure 1. Mechanism of diarrhea caused by RV infection. PLC -phospholipase C; 5-HT - 5-hydroxytryptamine, serotonin
Reviewer 2 Report
Comments and Suggestions for Authors
The manuscript entitled "Pathophysiological Pathways of Rotavirus Infections in Pediatric Patients: Understanding Disease Evolution and Associated Complications”, examines the key pathophysiological and immunological mechanisms driving disease progression and associated complications across different pediatric age groups. Despite the good structure of the manuscript, there are some concerns which should be addressed.
- Line 44; please correct “was observed” to be “revealed”.
- Please correct the order of section “1.1. Diarrhea and Vomiting” to be “3.1. Diarrhea and Vomiting”, and correct the subsequent sections order.
- Line 151; Correct this “RV only infected” to be “RV only infects”.
- Line 186 and 187; Correct this “presented in [Table 1]” to be presented in Table 1”.
- What is the difference between the * and ** in the Table 1.
- In table 1, Correct this “In encephalopathies of all kind” to be “In encephalopathies of all kinds”.
- Line 330; please add dot after [63].
- Line 335; Correct this “The [Table 2]” to be “The Table 2”.
- Line 339, same comment about Tble 3 as the previous comment.
- Please follow the same previous comment for all Tables mentioned throughout the whole manuscript.
- At the Vaccination section, please cite relevant publication “Insights into Recent Advancements in Human and Animal Rotavirus Vaccines: Exploring New Frontiers”, DOI: 10.1016/j.virs.2024.12.001.
- Line 513; please correct this “Table [Table 9]”
- The review lack any illustrative figures, I think you can add some attractive figures summarizing some sections.
- What are the limitations of the current diagnostic techniques for RV infections? Are there emerging methods that show promise for improved accuracy or speed?
Author Response
We would like to express our sincere gratitude for your thoughtful and insightful review. Your feedback has been immensely valuable, and we greatly appreciate your constructive suggestions. We remain fully open to any further comments or recommendations you may have.
In response to one of the reviews, we have decided to revise our manuscript by removing the section concerning diagnostics. The current version of our work focuses primarily on the pathophysiology of symptoms, treatment strategies, and rotavirus vaccination.
We have made a concerted effort to eliminate, as much as possible, references to narrative reviews. The revised manuscript now predominantly cites original research articles, cohort studies, case reports, guidelines issued by European scientific societies, meta-analyses, and systematic reviews.
- Line 44; please correct “was observed” to be “revealed”.
- The modification has been implemented.
2. Please correct the order of section “1.1. Diarrhea and Vomiting” to be “3.1. Diarrhea and Vomiting”, and correct the subsequent sections order.
- The modification has been implemented.
3. Line 151; Correct this “RV only infected” to be “RV only infects”.
- The modification has been implemented.
4. Line 186 and 187; Correct this “presented in [Table 1]” to be presented in Table 1”.
- The modification has been implemented.
5. What is the difference between the * and ** in the Table 1.
- The modification has been implemented.
6. In table 1, Correct this “In encephalopathies of all kind” to be “In encephalopathies of all kinds”.
- The modification has been implemented.
7. Line 330; please add a dot after [63].
- The modification has been implemented.
8. Line 335; Correct this “The [Table 2]” to be “The Table 2”.
- The modification has been implemented.
9. Line 339, same comment about Table 3 as the previous comment.
- The modification has been implemented.
10. Please follow the same previous comment for all Tables mentioned throughout the whole manuscript.
- The modification has been implemented.
11. In the Vaccination section, please cite the relevant publication “Insights into Recent Advancements in Human and Animal Rotavirus Vaccines: Exploring New Frontiers”, DOI: 10.1016/j.virs.2024.12.001.
- According to the suggestion of one of the reviewers, we have removed a significant part of the citations referring to review articles, focusing on experimental works. We have read the suggested article and will certainly cite it in our next work.
12. Line 513; please correct this “Table [Table 9]”
-
- The modification has been implemented.
13. The review lacks any illustrative figures, I think you can add some attractive figures summarizing some sections.
- The modification has been implemented.
- We have followed the recommendations and added one figure.
14. What are the limitations of the current diagnostic techniques for RV infections? Are there emerging methods that show promise for improved accuracy or speed
- At the suggestion of one of the reviewers, we have removed this part of the work. Thank you for your suggestion.
Reviewer 3 Report
Comments and Suggestions for Authors
The manuscript tiled “Pathophysiological Pathways of Rotavirus Infections in Pediatric Patients: Understanding Disease Evolution and Associated Complications” by Karolina et al., describes the pathophysilogical features of rotavirus infection in pediatric patients. The authors talked about local and systemic infection, disease progression, and clinical manifestation across various age groups. At the end the authors described the current strategies of diagnosis, treatment and vaccination based preventive measurements. The authors are advised to address following concerns:
1) The term “Pathways” in the title does not fit well. The review describes pathophysiological manifestation, progression, clinical features and treatment methodologies. Considering the overall text of the manuscript, ideal title could be “Pathophysiology of Rotavirus Infections in Pediatric Patients: Understanding Disease Evolution and Associated Complications”.
2) Replace RV with Rotavirus in keywords.
3) The section, “Materials and Methods” is not necessary for a review article. Although, the text included here is completely legitimate, this could be added as last paragraph in the “Introduction”.
4) Heading “AGE RV+ Progression - General Information” is confusing. Make it as “Disease progression across various age groups”.
Author Response
We would like to express our sincere gratitude for your thoughtful and insightful review. Your feedback has been immensely valuable, and we greatly appreciate your constructive suggestions. We remain fully open to any further comments or recommendations you may have.
In response to one of the reviews, we have decided to revise our manuscript by removing the section concerning diagnostics. The current version of our work focuses primarily on the pathophysiology of symptoms, treatment strategies, and rotavirus vaccination.
We have made a concerted effort to eliminate, as much as possible, references to narrative reviews. The revised manuscript now predominantly cites original research articles, cohort studies, case reports, guidelines issued by European scientific societies, meta-analyses, and systematic reviews.
1) The term “Pathways” in the title does not fit well. The review describes pathophysiological manifestation, progression, clinical features and treatment methodologies. Considering the overall text of the manuscript, the ideal title could be “Pathophysiology of Rotavirus Infections in Pediatric Patients: Understanding Disease Evolution and Associated Complications”.
- In response to your suggestions, as well as those of the other reviewers, we have decided to revise the title of the manuscript.
2) Replace RV with Rotavirus in keywords.
- The modification has been implemented.
3) The section, “Materials and Methods” is not necessary for a review article. Although, the text included here is completely legitimate, this could be added as last paragraph in the “Introduction”.
- The modification has been implemented.
4) Heading “AGE RV+ Progression - General Information” is confusing. Make it as “Disease progression across various age groups”.
- We agree with the suggestion and have implemented the change: 'AGE RV+ progression across various age groups'.
Reviewer 4 Report
Comments and Suggestions for Authors
Rotavirus Infections: Pathophysiology, Symptoms, Treatment and Vaccination
Rotavirus Infections: Pathophysiology, Symptoms, Treatment and Vaccination
Thank you for putting up this work that provides various aspects RV in a condensed form. i hav e a few comments and suggests.
Major comments:
- The review contains very useful information for the readers. All the aspects of RV are quite pertinent and to warrant this manuscript. However, there is need to realign the manuscript for in-depth discussion. This can be achieved through removal of redundant sections and focusing on a narrow path of RV. As noted, the vaccination is quite a huge section of the paper. In addition, there could be value in providing more indepth discussion on the various elements to avert the appearance of summaries of the RV subjects. This would avert the ending of section with the "need for more research". It would be help to have some theories and not so “obvious assertions” that are potentially contentious or untested before.
- The term Pathophysiology is appearing in the title and the only place where the word has been used. It is therefore not easy to understand the aspect of pathophysiology as it is mixed up in the other sections or indirectly implied. It would help to have this term used deliberately with prominence in the document as sign of sognificance. A look at the title/subheading can exemplify this observation.
- A glance at the subheading below shows some disparity between the title and the content. There appears to be headings which can safely be combined/condensed or omitted without much loss of information e.g., 2.2-2.4? and 3.1 and 3.2
- The tables and Figures do not seem to be adequately used in the Review to an extent that omission would not seriously affect the delivery. One can observe that there is barely a mention of the figure/Table without further elaboration or description. The suggestion is to use as much information from the table so that there is ample justification for the inclusion of the figures/tables in the body of the review.
- The conclusions of the review may not entirely be supported by the findings. It is suggested that authors pick out the key elements of the review and put them in the conclusion. For instance, the conclusion about “addressing the high dropout rate for vaccination” does not warrant to be part of the conclusion. It is a small part of the review that can be addressed in a minor way. This data on drop can easily be obtained from various sources within many public health institutions and should not be elevated to the level of conclusion c.f. the title. It would be beneficial to delve deep into matter associated with dropout. The manuscript should have an in-depth review of matters relating to the high dropout rate. As it is, the matter ends as a summary of high vaccination dropout rate. The suggestion would also include a critical reference to the title such that once the reader gets to the conclusion, the title is obviously justified.
- The review is expected to end in conclusion and perspective and theorization of ideas that stimulate the minds of readers to explore more of these avenues. After carrying out an in-depth review of literature the authors are allowed to come up with theories that push the boundaries of what has been accepted or normalized. It is from this end point that other researcher to pick out some loose ends to resolve the critical issues raised in the review.
Other comments:
Line 1: Although the study is mainly dealing with RV, it is apparent that the thrust is RV-A. This suggests that the title could be about RV-A rather than a general RV discussion.
Line 27; the key pathophysiologial and immunological mechanisms driving disease progression… are not clear in the document.
Line 46-50: There appears to be a contradiction as regards the increase in hospitalization but also the decrease due to high coverage.
Line 52: I suggest referring to the report as GBD study of 2019 (with the understanding the publication was in 2022 according to Ref No. 5)
Line 51: Could the authors explain how they arrived at this statement: …. “Overall, the global use of RV vaccines has the potential to prevent over one-third of all RV-related deaths.” I suggest there be a citation??
Line 59: spelling of “ut”
Line 83: This statement “extensive analysis” needs some clarity. It is not easy to find the aspects concerned with some of the mentions in the manuscript e.g., a look at section 2 does not seem to have a great deal of local infection as the rest of the section is about systemic infections as the subheading suggests.
Line 55-63: suggested that the paragraph moves up as paragraph 2 so that the readers are familiar with the virus by the time they get to read about the effects of the virus.
Line 79-82: This point has not been adequately addressed, as to how this study I going to have this global impact. Could the authors exemplify this assertion?
Line 97: The heading appears misplaced.
Line 98: AGE can be used here.
Line 108: I suggest this table be taken to the supplementary as there is little information that is extracted for the text to warrant its placement in the main body of the review. The only reference is that “the table provides a comparison”.
Line 114-129: The authors could bring out more information regarding this paragraph because the section still ends in some uncertainty about the two table. I suggest that the authors provide a possible a “universal scale” to the current situation.
Line 130: This section suggest that the focus is on RV-A. In addition, the local symptoms are not clearly outlined despite having a heading on “Local”.
Line 181: The section 2.2-2.7 should collapse into more focused and synthesized discussion.
Line 287-288: The authors could delve into more detail regarding the kind of research that us required to understand the impact.
Line 355-356: may require supporting information.
Line 379: there is need to more information on the cited table otherwise they may be placed as supplementary material or simply cited.
Line 387: spelling of Hydration
Line 491: Table 10 is too far from the text
Line 532-533: “the number of countries introducing vaccinations” is not evident from the graph.
Line 612: Clarify which section talks about NSP1 as the main protein responsible for evading the immune response
Line 555: Is this unique to RV vaccines?
Line 5.8: This section is redundant as there is not thorough explanation to it.
Line 624-262: Please clarify how this conclusion (particularly in Africa) was reached as the focus has been broadly worldwide with difference circumstances for specific regions
Some heading can be collapsed and others are redundant and could be safely removed. The headings can also be realigned to fit the title. It is also difficult to look at so many aspects of RV without
- Introduction
- Definition of Diarrhea and Scales for Its Assessment
- Local and Systemic Symptoms Caused by RV
2.1. Diarrhea and Vomiting
2.2. Systemic symptoms - General Information
2.3. Systemic Symptoms - Dehydration
2.4. Systemic Symptoms - RV Infections and CNS
2.5. RV and Autoimmunity
2.7 RV and Respiratory tract infections (RTIs)
- AGE RV+ progression across various age groups (2.5 pages)
3.1. Disease Progression in Children Under 5 Years of Age
3.2. Disease Progression in Children Over 5 Years of Age
- 4. Symptomatic treatment and antiviral therapy (2 pages)
4.1. Hydratation and Dietary management
4.2. Supplementary treatment
- Vaccination and Its Efficacy in Preventing RV Infections (7 pages)
5.1. The types of available vaccines with dosing schedules
5.2. Adverse effects associated with vaccine administration
5.3. Contraindications to vaccine administration
5.4. Effectiveness
5.5. Vaccination Coverage
5.6. Vaccination Dropout
5.7. RV Vaccines and Dilemmas: Vaccinating Immunocompromised Children, Premature Infants, and High Medical Risk Infants
5.8. Perspectives on next-generation RV vaccines
- Concluding Remarks and Perspective
Author Response
Dear Reviewer,
We would like to express our sincere gratitude for your thorough and constructive review of our manuscript. Your comments were extremely valuable and have contributed significantly to the revision and improvement of our work.
In response to your insightful comments, we have substantially revised and further refined the manuscript. The current version now places a greater focus on the pathophysiology of both local and systemic symptoms associated with RV infections. We have expanded the discussion to include proposed clinical scoring systems and added detailed information on the structure of the virus, emphasizing its relevance to the clinical presentation.
Moreover, we now explore in depth the differences in the course of infection in children under and over the age of five. Special attention has also been given to current and future vaccination strategies, including contraindications and existing challenges in vaccine implementation.
We hope that these revisions significantly improve the clarity and scope of the manuscript and that it now offers readers a comprehensive synthesis of the viral structure, associated symptoms, and underlying mechanisms, while also highlighting the essential role of vaccination in the prevention of RV infections.
Specifically:
- Redundancy and Focus: We carefully reviewed and streamlined the structure of the manuscript, removing or condensing sections that appeared redundant. As recommended, we have narrowed the scope of the paper to provide a more coherent and focused analysis of key aspects of RV infection, rather than offering a broad summary of the topic.
- Pathophysiology Emphasis: We took particular note of your comment regarding the use of the term pathophysiology. This concept now features more prominently throughout the text and is explicitly addressed in both the title and within clearly defined subheadings. We have enhanced the discussion to ensure that the pathophysiological mechanisms are distinctly presented and not merely implied within other sections.
- Section Reorganization: In light of your observation about certain subheadings (e.g., Sections 2.2–2.4 and 3.1–3.2), we have revised the section structure. Several headings have been combined or condensed to improve the flow of the manuscript and eliminate unnecessary repetition without compromising important content.
- Figures and Tables: We have revised the use of tables and figures to ensure they are fully integrated into the narrative. Each table and figure is now referenced and discussed in detail within the body of the manuscript, with clear justification for its inclusion. This allows the visual elements to meaningfully support the review rather than serve as mere supplementary materials. We also add new figures and tables.
- Conclusion Alignment: The conclusions have been thoroughly revised to more accurately reflect the key findings and discussions presented in the manuscript. The previous mention of vaccination dropout rates has been significantly reduced and is now addressed only briefly within the appropriate context. Instead, the conclusion now synthesizes the core insights of the review, aligns closely with the revised title, and highlights the most pertinent themes of the manuscript.
Other comments:
Line 1: Although the study is mainly dealing with RV, it is apparent that the thrust is RV-A. This suggests that the title could be about RV-A rather than a general RV discussion.
- The previous reviewer recommended incorporating references to RV-A where appropriate, and we have implemented these changes throughout the manuscript. The current emphasis on RV-A in some sections reflects this adjustment.
Line 27; the key pathophysiologial and immunological mechanisms driving disease progression… are not clear in the document.
- We have deliberately decided to focus the manuscript on the pathophysiology and symptomatology of RV infections, as well as vaccination strategies. Given the extensive scope of literature on immunological mechanisms, we opted to exclude a detailed discussion on immunology to maintain a manageable and coherent narrative.
Line 46-50: There appears to be a contradiction as regards the increase in hospitalization but also the decrease due to high coverage.
- We have revised this section to eliminate the noted contradiction and to clarify the relationship between hospitalization rates and vaccination coverage.
Line 52: I suggest referring to the report as GBD study of 2019 (with the understanding the publication was in 2022 according to Ref No. 5)
- The suggested clarification has been implemented; the report is now appropriately referred to as the "GBD Study 2019," acknowledging the publication year as 2022 in accordance with Reference No. 5.
Line 51: Could the authors explain how they arrived at this statement: …. “Overall, the global use of RV vaccines has the potential to prevent over one-third of all RV-related deaths.” I suggest there be a citation??
- The statement regarding the potential of RV vaccines to prevent a significant number of deaths has been modified and now includes an appropriate citation to support it.
Line 59: spelling of “ut”
- The spelling error has been corrected as advised.
Line 83: This statement “extensive analysis” needs some clarity. It is not easy to find the aspects concerned with some of the mentions in the manuscript e.g., a look at section 2 does not seem to have a great deal of local infection as the rest of the section is about systemic infections as the subheading suggests.
- The section has been reorganized and expanded to improve clarity. We have added content concerning both local and systemic infections, as well as potential mechanisms, to provide a more balanced and thorough analysis.
Line 55-63: suggested that the paragraph moves up as paragraph 2 so that the readers are familiar with the virus by the time they get to read about the effects of the virus.
- A new section has been created addressing viral structure, particularly focusing on proteins relevant to the pathophysiology and clinical manifestations of RV infections. This allows the reader to become familiar with the virus before progressing to its clinical effects.
Line 79-82: This point has not been adequately addressed, as to how this study I going to have this global impact. Could the authors exemplify this assertion?
- This part has been revised to better explain the intended global relevance and impact of the review. We have exemplified our statements with more specific arguments and references.
Line 97: The heading appears misplaced.
- The heading has been relocated and revised as recommended.
Line 98: AGE can be used here.
- We have adopted the suggested use of the abbreviation “AGE” in this section.
Line 108: I suggest this table be taken to the supplementary as there is little information that is extracted for the text to warrant its placement in the main body of the review. The only reference is that “the table provides a comparison”.
- We acknowledge the comment regarding Table placement. However, we believe that demonstrating the differences between scoring systems is important for understanding their development and potential refinement. Therefore, we have retained the table in the main text and expanded its contextual relevance.
Line 114-129: The authors could bring out more information regarding this paragraph because the section still ends in some uncertainty about the two table. I suggest that the authors provide a possible a “universal scale” to the current situation.
- We agree that further elaboration was needed. We have added additional discussion regarding the evaluation tools and emphasized the need for a more universal scoring system in the future.
Line 130: This section suggest that the focus is on RV-A. In addition, the local symptoms are not clearly outlined despite having a heading on “Local”.
- The section has been revised to more clearly outline local symptoms and to ensure consistency with the section headings. We have also reviewed the focus on RV-A to ensure it is balanced and appropriate.
Line 181: The section 2.2-2.7 should collapse into more focused and synthesized discussion.
- Sections 2.2 to 2.7 have been condensed and restructured to create a more focused and synthesized discussion, as recommended.
Line 287-288: The authors could delve into more detail regarding the kind of research that us required to understand the impact.
- We have expanded the discussion to include more detail on the types of future research needed to better understand the impact of RV infections and interventions.
Line 355-356: may require supporting information.
- Additional supporting information has been provided to strengthen the argument presented in this section.
Line 379: there is need to more information on the cited table otherwise they may be placed as supplementary material or simply cited.
- The relevant table has now been explicitly referenced and integrated into the body of the text to ensure its inclusion is justified.
Line 387: spelling of Hydration
- The spelling of "Hydration" has been corrected.
Line 491: Table 10 is too far from the text.
- We have repositioned Table 10 so that it appears closer to the relevant section of the text.
Line 532-533: “the number of countries introducing vaccinations” is not evident from the graph.
- We have revised the graphical representation and description to clarify the figure.
Line 612: Clarify which section talks about NSP1 as the main protein responsible for evading the immune response
- We delete it.
Line 555: Is this unique to RV vaccines?
Line 5.8: This section is redundant as there is not thorough explanation to it.
- The purpose of this section is to highlight ongoing research into new vaccine developments. We have expanded the section and added further explanation to underscore its relevance.
Line 624-262: Please clarify how this conclusion (particularly in Africa) was reached as the focus has been broadly worldwide with difference circumstances for specific regions
- The suggested clarification has been implemented.
We hope that these changes have significantly strengthened the manuscript and brought it in line with your expectations. Once again, we truly appreciate your detailed and thoughtful feedback, which has been instrumental in shaping the revised version of our review.
With kind regards,
The Authors
Reviewer 5 Report
Comments and Suggestions for Authors
The main problem with the manuscript is that the authors do not seem to see the portrait of the potential reader. The article does not contain any information that could be useful for clinicians, parents who decide whether to vaccinate or not, researchers who study this viral infection as part of basic research.
The abstract and introduction contain text that describes the content of the review in very general terms, it should be added how this review differs from a number of reviews previously published in other journals, and how it can be useful to the reader.
Other issues
1) The information presented in Table 5 is not suitable for a table, it is just a few lines of text, a bulleted list.
2) Captions to Fig. 1 and Fig. 2 should be below the figure
3) Too many references in lines 230, 232, 457, 458, etc.
4) Lines 368-372 don't contain any text
PS To make the review more useful to the reader and to collect many citations, it would be good to add one or more figures to the manuscript. The diagram shown in Figure 1 is not such a figure, but neither is Figure 2.
Comments on the Quality of English LanguageThe text of the manuscript will greatly benefit from being edited by a professional scientific editor.
Author Response
Dear Reviewer,
We sincerely thank you for your valuable feedback and for the time and effort you dedicated to reviewing our manuscript. We regret that the initial version did not meet your expectations, and we appreciate the opportunity to improve our work based on your thoughtful comments.
Following your suggestions, we have significantly revised and refined the manuscript. The current version now offers a more targeted focus on the pathophysiology of both local and systemic symptoms associated with RV infections. We have also expanded our discussion on proposed clinical scoring systems and incorporated detailed insights into the structure of the virus, highlighting its relevance to the clinical presentation of the disease.
Additionally, we now provide an in-depth comparison of the infection course in children under and over the age of five, and we have devoted substantial attention to current and future vaccination strategies, including contraindications and implementation challenges.
Importantly, in response to your critical observation regarding the lack of a clear target audience, we have revised the abstract and introduction to better define how this review differs from existing literature and who it aims to serve. Specifically:
- For clinicians, we offer a structured synthesis linking viral structure to symptomatology, highlight the limitations and progress of existing clinical scoring systems, and address practical challenges related to vaccination.
- For researchers, we integrate a detailed overview of current knowledge on RV-A pathogenesis and identify underexplored areas that warrant further investigation—especially concerning pathophysiological mechanisms and the need for standardized diagnostic criteria.
- For public health professionals and caregivers, we clarify the practical implications of vaccine coverage and dropout issues, and we aim to provide a balanced perspective that can support informed decision-making regarding immunization.
Other issues
1) The information presented in Table 5 is not suitable for a table, it is just a few lines of text, a bulleted list.
- The relevant table has now been explicitly referenced and integrated into the body of the text to ensure its inclusion is justified.
2) Captions to Fig. 1 and Fig. 2 should be below the figure
- We have implemented the suggested changes accordingly.
3) Too many references in lines 230, 232, 457, 458, etc.
- In response to comments from other reviewers, we acknowledge the necessity of including numerous citations to ensure transparency and to allow readers to access the original sources of the data we have used.
4) Lines 368-372 don't contain any text
- The section has been removed, as suggested.
PS To make the review more useful to the reader and to collect many citations, it would be good to add one or more figures to the manuscript. The diagram shown in Figure 1 is not such a figure, but neither is Figure 2.
- We have added new figures and tables to enhance the clarity and comprehensiveness of the manuscript.
Moreover, we have restructured parts of the abstract and introduction to clearly articulate the originality and relevance of our review in comparison with previously published works. We emphasize that this manuscript seeks to bridge the gap between molecular understanding, clinical presentation, and public health implications of RV infections—an approach we believe will provide added value to a broad readership.
We sincerely hope that these revisions have substantially improved the clarity, focus, and usefulness of our manuscript. Once again, thank you for your constructive and detailed feedback, which has greatly contributed to strengthening our review.
With kind regards,
The Authors
Round 2
Reviewer 1 Report
Comments and Suggestions for Authors
The introduction and abstract state that this review examines the key pathophysiological and immunological mechanisms driving disease progression and associated complications across different pediatric age groups. However, the manuscript does not adequately cover these objectives. Additionally, the claim in line 29 that "it evaluates current diagnostic techniques and therapeutic approaches" is not well-supported by the content. Taken together, I recommend rejecting the manuscript. My recommendation is to find an unique niche that has not been covered yet and I believe that the initial idea to analyze different age groups is worth analyzing.
Author Response
Dear Reviewer,
We sincerely thank you for your valuable feedback and the time you dedicated to reviewing our manuscript. We are sorry to learn that the initial version did not meet your expectations.
In response to your insightful comments, we have substantially revised and further refined the manuscript. The current version now places a greater focus on the pathophysiology of both local and systemic symptoms associated with RV infections. We have expanded the discussion to include proposed clinical scoring systems and added detailed information on the structure of the virus, emphasizing its relevance to the clinical presentation.
Moreover, we now explore in depth the differences in the course of infection in children under and over the age of five. Special attention has also been given to current and future vaccination strategies, including contraindications and existing challenges in vaccine implementation.
We hope that these revisions significantly improve the clarity and scope of the manuscript and that it now offers readers a comprehensive synthesis of the viral structure, associated symptoms, and underlying mechanisms, while also highlighting the essential role of vaccination in the prevention of RV infections.
Once again, we are grateful for your constructive input, which has greatly contributed to the improvement of our work.
With kind regards,
The Authors
Reviewer 4 Report
Comments and Suggestions for Authors
Dear Authors,
Thank you for extensively addressing the comments. Your effort is greatly appreciated.
Author Response
We are very grateful for the reviewer's kindness and suggestions.
Reviewer 5 Report
Comments and Suggestions for Authors
The manuscript has been significantly improved according to the reviewers' comments. But the use of Refs as in lines 115, 116, 302, 304, 469, 470 should be avoided
Comments on the Quality of English LanguageUse of professional scholar English editing services is recommended
Author Response
Dear Reviewer,
We would like to express our sincere gratitude for your valuable and insightful comments regarding both the quality of the English language and the use of references in our manuscript. Your careful review and thoughtful suggestions have significantly contributed to enhancing the quality of our work.
In response to your recommendation concerning the English language, we have undertaken a comprehensive revision of the manuscript to improve its clarity, style, and grammatical accuracy.
Additionally, in response to your suggestion regarding the presentation of references, we have revised the formatting to improve the readability and coherence of the manuscript. Specifically, rather than listing references individually (e.g., [10, 11, 12, 13, 14, 15, 16, 17, 18]), we have consolidated them into a more compact range format (e.g., [10-18]). We also wish to respectfully clarify that the references cited in these instances are indispensable for maintaining the scientific accuracy and completeness of the manuscript:
- Lines 115 and 116: The cited references are critical for the accurate description of the structure of the rotavirus (RV) and the specific roles of its proteins. To ensure precision and avoid reliance on secondary review articles, as suggested by other reviewers, we have prioritized citing original research studies.
- Lines 302 and 304: The various clinical manifestations of central nervous system (CNS) involvement caused by RVs are highly complex and multifaceted. Summarizing this information solely based on general reviews would, in our view, compromise the depth and scientific rigor of the manuscript.
- Lines 469 and 470: Each vaccine formulation against RV is supported by distinct and independent studies. Retaining separate references ensures that the unique characteristics and evaluation outcomes of each vaccine are accurately acknowledged, thus maintaining the manuscript's scientific integrity.
We sincerely appreciate your thoughtful feedback and your meticulous attention to detail, which have greatly assisted us in improving the overall quality and clarity of our manuscript.
Round 3
Reviewer 1 Report
Comments and Suggestions for Authors
All the major comments have been addressed. However, there are few minor comments remain to be addressed:
- Make sure that all abbreviations are spelled out in full when first introduced (e.g., Rotavirus [RV]). After that, there is no need to spell them out again.
- Make sure that the abstract and conclusion remarks reflect the manuscript content.
Author Response
Dear Reviewer,
We would like to extend our sincere thanks for your thoughtful and constructive feedback. We have
carefully addressed the remaining minor comments as follows:
All abbreviations are now spelled out in full upon their first appearance in the manuscript (e.g.,
Rotavirus [RV]), in accordance with your recommendation.
We have also revised the abstract and the concluding remarks to ensure they accurately and
clearly reflect the content and key findings of the manuscript.
We are grateful for your valuable suggestions, which have significantly contributed to enhancing the
clarity and overall quality of our work.
With kind regards,
Authors